# Concentration of Zearalenone, Alpha-Zearalenol and Beta-Zearalenol in the Myocardium and the Results of Isometric Analyses of the Coronary Artery in Prepubertal Gilts

**DOI:** 10.3390/toxins13060396

**Published:** 2021-06-02

**Authors:** Magdalena Gajęcka, Michał S. Majewski, Łukasz Zielonka, Waldemar Grzegorzewski, Ewa Onyszek, Sylwia Lisieska-Żołnierczyk, Jerzy Juśkiewicz, Andrzej Babuchowski, Maciej T. Gajęcki

**Affiliations:** 1Department of Veterinary Prevention and Feed Hygiene, Faculty of Veterinary Medicine, University of Warmia and Mazury in Olsztyn, Oczapowskiego 13/29, 10-718 Olsztyn, Poland; lukaszz@uwm.edu.pl (Ł.Z.); gajecki@uwm.edu.pl (M.T.G.); 2Department of Pharmacology and Toxicology, Faculty of Medical Sciences, University of Warmia and Mazury in Olsztyn, Warszawska 30, 10-082 Olsztyn, Poland; michal.majewski@uwm.edu.pl; 3Institute of Biology and Biotechnology, College of Natural Sciences, University of Rzeszów, Pigonia 1, 35-310 Rzeszow, Poland; wgrzegorzewski@ur.edu.pl; 4Interdisciplinary Center for Preclinical and Clinical Research, Department of Biotechnology, Institute of Biol-ogy and Biotechnology, College of Natural Sciences, University of Rzeszów, Pigonia 1, 35-310 Rzeszow, Po-land; 5Dairy Industry Innovation Institute Ltd., Kormoranów 1, 11-700 Mrągowo, Poland; ewa.onyszek@iipm.pl (E.O.); andrzej.babuchowski@iipm.pl (A.B.); 6Independent Public Health Care Centre of the Ministry of the Interior and Administration, and the Warmia and Mazury Oncology Centre in Olsztyn, Wojska Polskiego 37, 10-228 Olsztyn, Poland; lisieska@wp.pl; 7Department of Biological Function of Foods, Institute of Animal Reproduction and Food Research, Division of Food Science, Tuwima 10, 10-748 Olsztyn, Poland; j.juskiewicz@pan.olsztyn.pl

**Keywords:** zearalenone, low doses, carry-over, myocardium, vascular reactivity

## Abstract

The carry-over of zearalenone (ZEN) to the myocardium and its effects on coronary vascular reactivity in vivo have not been addressed in the literature to date. Therefore, the objective of this study was to verify the hypothesis that low ZEN doses (MABEL, NOAEL and LOAEL) administered per os to prepubertal gilts for 21 days affect the accumulation of ZEN, α-ZEL and β-ZEL in the myocardium and the reactivity of the porcine coronary arteries to vasoconstrictors: acetylcholine, potassium chloride and vasodilator sodium nitroprusside. The contractile response to acetylcholine in the presence of a cyclooxygenase (COX) inhibitor, indomethacin and / or an endothelial nitric oxide synthase (e-NOS) inhibitor, L-NAME was also studied. The results of this study indicate that the carry-over of ZEN and its metabolites to the myocardium is a highly individualized process that occurs even at very low mycotoxin concentrations. The concentrations of the accumulated ZEN metabolites are inversely proportional to each other due to biotransformation processes. The levels of vasoconstrictors, acetylcholine and potassium chloride, were examined in the left anterior descending branch of the porcine coronary artery after oral administration of ZEN. The LOAEL dose clearly decreased vasoconstriction in response to both potassium chloride and acetylcholine (*P* < 0.05 for all values) and increased vasodilation in the presence of sodium nitroprusside (*P =* 0.021). The NOAEL dose significantly increased vasoconstriction caused by acetylcholine (*P* < 0.04), whereas the MABEL dose did not cause significant changes in the vascular response. Unlike higher doses of ZEN, 5 μg/kg had no negative influence on the vascular system.

## 1. Introduction

Zearalenone and its metabolites, á-zearalenol (á-ZEL) and â-zearalenol (â-ZEL), are commonly encountered in plant materials [1]. Since their structure resembles that of estradiol, mycotoxins contribute to reproductive disorders [2,3,4]. Alpha-zearalenol is the main ZEN metabolite that affects pigs. Other animal species (such as broiler chickens, cows and sheep) are more susceptible to β-ZEL whose metabolic activity is lower [5]. The activity of ZEN is determined by biotransformation processes in plants [6] and animals [7], the immune status [8,9,10] of the reproductive system (during puberty, reproductive cycle and pregnancy—due to changes in steroid hormones concentrations) [11,12,13] and the digestive system of animals exposed to this mycotoxin [3,4,14,15].

Different doses of mycotoxins exert various effects [4,15]. Both the symptoms and health (toxicological) effects of high doses of most mycotoxins have been extensively studied [16]. Animals can tolerate long-term exposure to low monotonic doses of mycotoxins [3,17] which actually may serve vital life needs [9,18,19,20,21] or therapeutic purposes [22]. However, low mycotoxin doses may also have a negative impact on health [23]. A low dose [24] was defined in our previous clinical studies [15,25,26] based on the presence or absence of clinical symptoms of ongoing ZEN mycotoxicosis (e.g., changes in estradiol, progesterone or testosterone levels [4], changes in blood biochemical parameters and body weight [15,17] or quantitative and qualitative changes in the intestinal microbiome [3,26]). Three doses were identified based on the results of our previous research and other authors’ findings: the lowest observed adverse effect level (LOAEL, >10 μg ZEN/kg BW) [3,12,27] dose which causes clinical symptoms [28]; the highest no observed adverse effect level (NOAEL, 10 μg ZEN/kg BW) [29] dose, also referred to as the maximum safe starting dose [24], which does not cause clinical symptoms (subclinical states); and the minimal anticipated biological effect level (MABEL, <10 μg ZEN/kg BW) dose, namely the lowest measurable dose [24] or the effective in vivo dose that positively interacts with the host organism in various stages of life, and produces measurable effects without any side effects [24].

The dose-response relationship has been undermined by the low-dose hypothesis, especially with respect to chemical compounds exhibiting hormonal activity [30] such as ZEN-type mycoestrogen and its metabolites, which are endocrine disruptors (EDs) even when administered in low doses [13]. The dose-response relationship does not permit a direct analysis / meta-analysis of the risk (clinical symptoms or the results of laboratory tests) resulting from the transition from high to low doses [31]. The concept of the lowest identifiable dose, namely a dose that produces an effect contrary to the expected outcome, is gaining increasing popularity in biomedical sciences. The associated mechanisms have to be investigated to support decision making in selected processes [15,32]. 

Substances that can contribute to both maintaining and disrupting homeostasis have challenged the traditional concepts in toxicology, particularly “the dose makes the poison” adage. Zearalenone and its metabolites (ZELs) have been found to evoke different responses when administered in low doses to mammals [3], which has also been reported by Knutsen et al. [23]. According to the Scientific Panel on Contaminants in the Food Chain (CONTAM), the impact of ZEN on the health status of animals needs to re-evaluated, taking into account the responses of different animal species to the lowest detectable doses of ZEN (LOAEL, NOAEL, MABEL) [23,33,34] found in feed, including both the parent compound and its derivatives [15]. 

Our previous research revealed that ZEN accelerates eryptosis, namely the apoptosis of red blood cells. Due to its mechanism of action [15,35], ZEN increases intracellular Ca^2+^ [35,36,37] levels, induces oxidative stress and decreases energy resources [38]. These responses are linked with the pathogenesis of anemia and microcirculatory disorders. Microcirculatory disorders are observed in various tissues during exposure to ZEN, as manifested by numerous extravasations and the presence of vessels with dilated lumina and multiple extravasations [20,39] in prepubertal female mammals. The observed changes prompted the hypothesis that ZEN and/or its metabolites can act as vasodilators in the presence of vasoconstrictors [40,41]. Therefore, the objective of this study was to determine whether exposure to low doses of ZEN (MABEL, NOAEL—the highest dose and LOAEL—one of the lowest doses) administered per os to prepubertal gilts for 21 days reach the myocardium and induces changes in reactivity in response to vasoconstrictors (acetylcholine and potassium chloride) and vasodilators (sodium nitroprusside) in the left anterior descending branch of the coronary artery. 

## 2. Results

### 2.1. Experimental Feed

The feed analyzed in this experiment did not contain any mycotoxins, or its mycotoxin content was below the sensitivity of the method (VBS). The concentrations of modified and masked mycotoxins were not analyzed.

### 2.2. Clinical Observations 

Experimental animals did not exhibit the clinical signs of ZEN mycotoxicosis were not observed during the experiment. However, changes in specific tissues or cells were frequently noted in analyses of selected serum biochemical profiles, genotoxicity of cecal water, selected steroid concentrations and intestinal microbiota parameters in samples collected from the same animals and in analyses of the animals’ growth performance. The results of these analyses were presented in our previous studies [3,4,15,25,26]. 

### 2.3. Concentrations of Zearalenone and Its Metabolites in the Heart Muscle 

In general, the concentrations of ZEN and its metabolites in the myocardium of prepubertal gilts did not differ significantly between analytical dates or experimental groups (Table 1).

Highly significant difference in the concentrations of β-ZEL was noted on D2 (exposure day 21) between group E1 (5 μg ZEN/kg BW) (0.204 ng/g—highest value) vs. groups E2 (10 μg ZEN/kg BW) and E3 (15 μg ZEN/kg BW) (difference of 0.127 and 0.172 ng/g, respectively). On the remaining days of the experiment, relatively high but not statistically significant differences were observed between mean values ($\bar{x}$). On D1 (exposure day 7), ZEN levels were lowest in group E1 and highest in E2, i.e., they were inversely proportional to the administered dose. The concentrations of ZEN metabolites were proportional to the applied dose. The proportionality of ZEN and α-ZEL concentrations relative to the administered dose was maintained in the apex of the heart on D2. In contrast, β-ZEL concentrations were highest in group E1, lower in group E2 and lowest in group E3.

#### Carry-Over Factor

The carry-over factor (CF) was calculated to determine the release of ZEN and its metabolites from the gastrointestinal tract and their absorption and, possibly, distribution [4] to, e.g., the myocardium of prepubertal gilts. The CF values for ZEN (Table 1) in the myocardium were determined in the range of 8 × 10^−6^ in group E1 on D1 to 16 × 10^−6^ in group E3 on D2. The CF values for α-ZEL ranged from 0.0 in group E1 to 14 × 10^−7^ in group E2 on D1, and from 1 × 10^−6^ in groups E2 and E3 to 14 × 10^−7^ in group E1 on D2 (ZEN 5). The CF for β-ZEL ranged from 0.0 in group E1 to 2 × 10^−7^ in group E2 on D1, and from 1 × 10^−7^ in group E3 to 2 × 10^−6^ in group E1 on D2. These values were proportionally lower than the values noted in other tissues (not in the heart muscle) during exposure to higher doses of ZEN [17,21,42,43,44].

### 2.4. Vascular Reactivity Analyses

Porcine coronary arteries (PCAs) contracted in response to KCl within the concentration range of 2.5 to 30 mM (Figure 1A–D). The calculated D1_AUC_ values were: $\frac{E1}{control}=1.61,\;P<0.05$; $\frac{E2}{control}=1.84,\;P<0.01$; $\frac{E3}{control}=0.54,\;P<0.001$, meanwhile D2_AUC_ ratio was: $\frac{E1}{control}=1.38,\;P=0.6,\;\frac{E2}{control}=1.00,\;P=0.9,\frac{E3}{control}=0.39,\;P<0.001$ (Figure 1B). A significant difference in the contractile response on $\frac{D2}{D1}AUC$ was noted only in E2 vessels (0.53-fold, *P* < 0.01), but not in the control (0.97-fold, *P* = 0.8), E1 (0.83-fold, *P* = 0.6) and E3 (0.70-fold, *P* = 0.15).

Acetylcholine concentrations of 10^−7^ to 10^−5^ M induced significant contraction of porcine coronary arteries (Figure 2A–D and Table 2). The calculated D1_AUC_ values were based on the AUC (Figure 2A): $\frac{E1}{control}=0.57,\;P<0.01$; $\frac{E2}{control}=1.33,\;P<0.05$; $\frac{E3}{control}=0.27,\;P<0.001$ and D2_AUC_: $\frac{E1}{control}=0.49,\;P<0.001$; $\frac{E2}{control}=1.15,\;P<0.6$; $\frac{E3}{control}=0.25,\;P<0.001$ (Figure 2B). No significant differences in the $\frac{D2}{D1}AUC$ ratio was observed on the control (0.98-fold), E1 (0.83-fold), E2 (0.85-fold) and E3 (0.91-fold), all *P* values ≥ 0.8.

Sodium nitroprusside (10^−7^–10^−4^ M) caused a concentration-dependent relaxation of PCAs (Figure 3A,B and Table 2), with the onset at 10^−7^ M and maximal response at 10^−4^ M (Figure 3C,D). The calculated D1_AUC_ values were: $\frac{E1}{control}=0.85,\;P=0.5$; $\frac{E2}{control}=0.94,\;P=0.8$; $\frac{E3}{control}=1.03,\;P=0.9$ D2_AUC_: $\frac{E1}{control}=0.73,\;P<0.05$; $\frac{E2}{control}=1.09,\;P=0.8$; $\frac{E3}{control}=1.39,\;P<0.01$ (Figure 3B). A significant difference in the $\frac{D2}{D1}AUC$ was noted in E1 (0.64-fold, *P* < 0.05). This was not observed for the control (0.74-fold, *P* = 0.6), E2 (0.86-fold, *P* = 0.8) and E3 (0.948-fold, *P* = 0.9). 

The calculated AUC, Emax (%) and pD2 for acetylcholine and sodium nitroprusside are presented in Table 2.

Neither L-NAME (4.4 × 10^−5^ M) nor indomethacin (4.4 × 10^−6^ M) had a significant effect on baseline vascular tone (data not presented). Indomethacin did not affect arterial sensitivity to acetylcholine in control PCA (0.80, *P* > 0.3), but it decreased the response in E1 (0.55, *P* < 0.001) and E2 (0.74, *P* < 0.01), and potentiated the response in E3 (2.39, *P* < 0.0001; Figure 4). L-NAME increased the sensitivity of the PCA to acetylcholine in all studied groups: control (1.36, *P* < 0.05), E1 (2.48, *P* < 0.001), E2 (1.29, *P* < 0.01) and E3 (3.39, *P* < 0.001). Only in E3, preincubation with both indomethacin + L-NAME increased arterial sensitivity to acetylcholine 5.11-fold (*P* < 0.001) vs. control conditions. The above was not observed in the control group (1.06, *P* = 0.3), E1 (0.87, *P* = 0.8) or group E2 (1.22, *P* = 0.4).

## 3. Discussion

An analysis of the physiological condition of prepubertal gilts indicates that ZEN acts as both an undesirable substance and an endocrine disruptor (ED) [4]. Even when ingested at MABEL, NOAEL (highest) and LOAEL (very low) doses, ZEN significantly increases the concentrations of selected hormones and causes hyperestrogenism, i.e., supraphysiological hormone levels [4,15,42], in prepubertal gilts. Zearalenone is also characterized by a non-monotonic dose-response curve (according to the principle of hormesis [43]). Therefore, the results of research studies investigating the effects of different ZEN doses on tissues [44,45], cells [46] and cell organelles [47] are difficult to compare. 

### 3.1. Zearalenone and Its Metabolites in the Heart Muscle

In the present study, the carry-over of ZEN and its metabolites in the myocardium of prepubertal gilts was highly individualized (absence of significant differences due to high variation in SD values) (Table 1). The presence of ZEN and a steady increase in its concentrations, proportional to the administered dose, were noted in the myocardium of gilts in groups E1 and E2 on D1. Zearalenone levels were much lower in group E3 on D1, which is partially consistent with previous findings [44]. Similar conclusions were drawn by Gajęcka et al. [17] from a study of female wild boars. The concentrations of α-ZEL (rising trend) and β-ZEL in the myocardium were inversely proportional to each other, which, in our opinion, is a normal response [4]. The bioavailability of ZEN and its metabolites in the myocardium is affected by biotransformation processes in prepubertal females. Interestingly, the distribution of ZEN and metabolite concentrations in the myocardium was similar to the values reported in blood by Rykaczewska et al. [4]. In contrast to the results reported by Yan et al. [44], ZEN metabolites were not detected on D1 (or were below the sensitivity of the method), which could be due to the low supply of endogenous steroid hormones. According to other studies [48,49], a deficiency of ovarian hormones in mammals leads to pressure overload, thus compromising cardiac function. Supplementation with 17β-estradiol [50] or mycoestrogen can reverse these effects or alter the profile of estrogen hormones (by modulating feminization) [4,51]. It should also be stressed that the 7th day of exposure (D1) marks the end of adaptive processes, in particular adaptive immunity [52]. These substances could also be used as substrates that regulate the expression of genes encoding hydroxysteroid dehydrogenase [3], a molecular switch that enables the modulation of steroid hormone prereceptors. These processes were most visible in group E1, where only the parent mycotoxin was detected (100%). In the remaining groups, the presence of metabolites was noted, and their concentrations increased proportionally to the applied dose. The observations made in group E1 (MABEL dose) indicate that prepubertal females utilize even the smallest amounts of estrogen-like substances (what are they zearalenone and its metabolites—[44]) to compensate for endogenous estrogen deficiency (inducing supraphysiological hormonal levels in prepubertal females—[4]), which can increase cardiac automaticity [53].

On D2, ZEN concentrations increased proportionally to the administered dose and were higher than on D1. In group E1, the proportions of both metabolites (%) were higher than in groups E2 and E3 (group E1: ZEN—73.62%, α-ZEL—11%, β-ZEL—15.37%; group E2: 91.48%, 5.82% and 2.68%, respectively; group E3: 93.42%, 5.96% and 0.61%, respectively). Similar to D1, the concentrations of α-ZEL (rising trend) and β-ZEL in the myocardium were inversely proportional to each other. The levels of α-ZEL were higher, whereas β-ZEL levels were lower in groups E2 and E3. This could result from the saturation of myocardial tissue with ZEN and its metabolites, e.g., active estrogen receptors [54] as well as other factors that influence the demand for ZEN and ZEN-like mycotoxins over time of exposure [55]. Unlike in the current experiment, Yan et al. [44] did not detect ZEN, but identified both ZEN metabolites in samples of heart muscle tissue. However, the cited study was conducted in vitro, and the animals’ age or sex were not specified, which makes it impossible to directly compare the above results with our findings. 

The carry-over of ZEN, an exogenous estrogen-like substance, from the porcine gastrointestinal tract to myocardial tissue via the blood was also analyzed by calculating the CF. The CF values for myocardial tissue in prepubertal gilts have never been determined in the literature, in particular during exposure to three low, monotonic doses of ZEN for 21 consecutive days. Even a cursory analysis of CF values indicates that the accumulation of ZEN and its metabolites was much lower in the myocardium than in the blood [4]. Mycotoxin concentrations ranged from 1 × 10^−1^ to 1 × 10^−3^ in the blood, and from 0 (only in group E1 on D1 for both metabolites) to 1 × 10^−7^ (in the remaining groups on both D1 and D2) in the myocardium. These observations suggest that differences in carry-over decrease the accumulation of ZEN and its metabolites in the myocardium [48]. These differences are very difficult to explain. Based on the existing knowledge and the extrapolation of previous results, it could be suggested that by disrupting endocrine processes, EDs exert specific effects on cells and tissues [4,51,56] and modulate the structure and functions of the heart muscle [48]. Most importantly, EDs can induce different responses, depending on the dose, exposure duration and the stage of growth and development in mammals [15], in particular females.

Therefore, it can be hypothesized that low doses of undesirable substances (including ZEN) exert minor or much smaller effects on myocardial homeostasis, compared with other cells and tissues in the studied animals due to much lower availability. 

### 3.2. Isometric Tension Analyses

The vasodilatory and vasoconstrictive properties of isolated porcine coronary arteries with an intact endothelium, which regulate vascular smooth muscle contraction, were also analyzed in the study. The blood vessels in various organs and species may respond differently to agonists and antagonists. Potassium chloride and acetylcholine induce vasocontraction, whereas sodium nitroprusside induces vasodilation in porcine coronary arteries (PCAs). 

The KCl-induced contraction of the PCAs was enhanced in groups E1 and E2 on D1. However, a decreased response was noted in group E3. On D2, KCl-induced vasoconstriction did not differ in groups E1 and E2, but it decreased further in group E3. These results indicate that the sensitivity of smooth muscles of PCAs to K^+^ is highly dependent on the concentrations of ZEN in the diet and the duration of exposure to this mycotoxin.

Acetylcholine-induced contraction decreased in group E3, which is similar to the response observed for KCl, so this effect might not be entirely dependent on the muscarinic receptors. Surprisingly, decreased response was also observed in group E1 but not in group E2. Acetylcholine’s effect on vascular tension is dependent on muscarinic receptors [57], which suggests that ZEN is able to modulate the function of these receptors.

Sodium nitroprusside is a donor of exogenous nitric oxide with the endothelium-independent effect. In this study, the sensitivity of PCAs to the nitric oxide was increased in group E3 after prolonged exposure. Surprisingly, arterial sensitivity to nitric oxide decreased in group E1, which suggests that the sensitivity of smooth muscles to nitric oxide changes in response to dietary ZEN, which is endothelium-independent mechanism. These results also indicate that smooth muscles of PCA are targeted by ZEN and its metabolites, and that ZEN my regulate the mechanism(s) of nitric oxide synthesis, which is dose- and time-dependent. Further investigation is needed to examine the mechanism(s) underlying different responses to ZEN and their potential dependence on the endothelium. 

The analysis of the effects of COX and e-NOS inhibitors shed a new light on the properties of ZEN. COX inhibitors potentiated ACh-induced vasoconstriction only in group E3. This response decreased in groups E1 and E2, whereas no significant changes were found in the control group. These findings suggest that ACh-induced vasoconstriction in group E3 was at least partly dependent on the net vasodilator effect of prostanoids, whereas the decreased response in groups E1 and E2 was dependent on the vasoconstrictor effect of prostanoids. e-NOS inhibitors increased vasoconstriction in all groups (C, E1, E2 and E3), which indicates that nitric oxide plays a key role in vascular tone regulation of PCA. However, this effect was more pronounced in groups E3 (3.39-fold) and E1 (2.48-fold) than in group E2 (1.29-fold) and the control (1.36-fold), which suggests that ZEN is able to modulate the bioavailability or sensitivity of nitric oxide. When both COX and e-NOS are blocked, mechanisms other than prostanoids and nitric oxide are engaged in vascular tone regulation. These mechanisms are regulated by hormonal changes and, possibly, ZEN. Acetylcholine’s effects were potentiated in the presence of COX and e-NOS inhibitors in group E3, but not in groups E1 or E2. 

These results indicate that a different mechanism is responsible for the net vasoconstrictor effect which was upregulated only in group E3. The vasodilator effect of prostanoids (PGI_2_) and nitric oxide was upregulated by the administered ZEN dose. The endothelium-derived hyperpolarizing factor (EDHF) could be yet another mechanism of vascular control. The major routes of EDHF regulation include the metabolism of arachidonic acid to epoxyeicosatrienoic acids (EETs), potassium channels, gap junctions and hydrogen peroxide [58]. However, further research is needed to clarify the exact mechanism(s), including EDHF, by which ZEN acts on PCA.

### 3.3. Conclusions

The following conclusion can be drawn from the present study: among ZEN doses analyzed both in vivo and in vitro, the presence of ZEN and its metabolites in the myocardium is found even at the MABEL dose and could be a safe dose for the myocardium regardless of the time of exposure. Meanwhile LOAEL highly affects the functioning of the porcine coronary arteries.

## 4. Materials and Methods 

### 4.1. In Vivo Study 

#### 4.1.1. General Information

All experimental procedures involving animals were carried out in compliance with Polish regulations setting forth the terms and conditions of animal experimentation (Opinions No. 12/2016 and 45/2016/DLZ of the Local Ethics Committee for Animal Experimentation of 27 April 2016 and 30 November 2016).

#### 4.1.2. Experimental Animals and Feed

The in vivo experiment was conducted at the Department of Veterinary Prevention and Feed Hygiene of the Faculty of Veterinary Medicine at the University of Warmia and Mazury in Olsztyn on 40 clinically healthy prepubertal gilts with initial BW of 14.5 ± 2 kg. The animals were housed in pens, and they had free access to water. Throughout the experiment, gilts in all groups received the same feed. The animals were randomly divided into three experimental groups (group E1, group E2 and group E3; n = 10) and a control group (group C, n = 10). Group E1 gilts were orally administered ZEN (SIGMA-ALDRICH Z2125-26MG USA) at 5 μg ZEN/kg BW, group E2 gilts received 10 μg ZEN/kg BW and group E3 gilts—15 μg ZEN/kg BW. 

Analytical samples of ZEN were dissolved in 96 µl of 96% ethanol (SWW 2442-90, Polskie Odczynniki SA, Poland) in doses appropriate for different BW. Feed saturated with different doses of ZEN in an alcohol solution was placed in gel capsules. Before administration to the animals, the capsules were stored at room temperature before administration to evaporate the alcohol. In the experimental groups, ZEN was administered daily in gel capsules before morning feeding. The animals were weighed every week, and the results were used to adjust mycotoxin doses on an individual basis. Feed was the carrier, and control group gilts received the same gel capsules, but without mycotoxins. 

The feed offered to all groups of experimental animals was supplied by the same producer. Throughout the experiment, feed was provided ad libitum in loose form, twice daily (at 8:00 a.m. and 5:00 p.m.). The manufacturer’s declared composition of the complete diet is shown in Table 3. 

The proximate chemical composition of the diets fed to gilts in groups C, E1, E2 and E3 was evaluated with the use of the NIRS™ DS2500 F feed analyzer (FOSS, Hillerød, Denmark) which is a monochromator-based NIR reflectance and transflectance analyzer with a scanning range of 850–2500 nm.

#### 4.1.3. Toxicological Analysis of Feed

Feed was analyzed for the presence of ZEN and DON. Mycotoxin content was determined by extraction on immunoaffinity columns (Zearala-TestTM Zearalenone Testing System, G1012, VICAM, Watertown, MA, USA; DON-TestTM DON Testing System, VICAM, Watertown, MA, USA) and in a high-performance liquid chromatography (HPLC) system (Hewlett Packard type 1100 and 1260) with a mass spectrometer (MS) and a chromatography column (Atlantis T3 3 μm 3.0 × 150 mm Column No. 186003723, Waters, AN Etten-Leur, Ireland). The mobile phase was an 80:10 mixture of water and acetonitrile with the addition of 2 mL of acetic acid per 1 L of the mix. The flow rate was 0.4 mL/min. The obtained values did not exceed the limit of quantification (LoQ) set at 2 ng/g for ZEN and 5 ng/g for DON. The analyzed compounds were quantified at the Department of Veterinary Prevention and Feed Hygiene, Faculty of Veterinary Medicine, University of Warmia and Mazury in Olsztyn [14].

#### 4.1.4. Toxicological Analysis of the Apex of the Heart

##### Tissues Samples

Five prepubertal gilts from every group were euthanized on analytical dates 1 (D1-exposure day 7) and 2 (D2—exposure day 21) by intravenous administration of pentobarbital sodium (Fatro, Ozzano Emilia BO, Italy) and bleeding. Samples were collected from the myocardium (the apex of the heart) immediately after cardiac arrest and were rinsed with phosphate buffer. The collected samples were stored at a temperature of −20 °C. 

##### Extraction Procedure

The presence of ZEN, α-ZEL and β-ZEL in the apex of the heart was determined with the use of immunoaffinity columns (Zearala-TestTM Zearalenone Testing System, G1012, VICAM, Watertown, MA, USA). All extraction procedures were carried out in accordance with the recommendations of column manufacturers. After extraction, the eluents were placed in a water bath at a temperature of 50 °C and were evaporated in a stream of nitrogen. Dry residues were stored at −20 °C until chromatographic analysis. Next, 0.5 mL of 99% acetonitrile (ACN) was added to dry residues to dissolve the mycotoxin. The process was monitored with the use of internal standards, and the results were validated by mass spectrometry.

##### Chromatographic Quantification of ZEN and Its Metabolites

Zearalenone and its metabolites were quantified at the Institute of Dairy Industry Innovation in Mrągowo. The biological activity of ZEN, α-ZEL and β-ZEL in the myocardium was determined by combined separation methods, immunoaffinity chromatography (Zearala-TestTM Zearalenone Testing System, G1012, VICAM, Watertown, MA, USA), liquid chromatography (LC) (Agilent 1260 LC system) and mass spectrometry (MS). Samples were analyzed on a chromatographic column (Atlantis T3, 3 μm 3.0 × 150 mm, column No. 186003723, Waters, AN Etten-Leur, Ireland). The mobile phase was composed of 70% acetonitrile (LiChrosolvTM, No. 984 730 109, Merck-Hitachi, Mannheim, Germany), 20% methanol (LiChrosolvTM, No. 1.06 007, Merck-Hitachi, Mannheim, Germany) and 10% deionized water (MiliporeWater Purification System, Millipore S.A. Molsheim-France) with the addition of 2 mL of acetic acid per 1 L of the mixture. The column was flushed with 99.8% methanol (LIChrosolvTM, No. 1.06 007, Merck-Hitachi, Mannheim, Germany) to remove the bound mycotoxin. The eluents were placed in a water bath with a temperature of 50 °C, and the solvent was evaporated in a stream of nitrogen. Mycotoxin concentrations were determined with an external standard and were expressed in ppb (ng/mL). Matrix-matched calibration standards were applied in the quantification process to eliminate matrix effects that can decrease sensitivity. Calibration standards were dissolved in matrix samples based on the procedure that was used to prepare the remaining samples. The material for calibration standards was free of mycotoxins. The limits of detection (LOD) for ZEN, α-ZEL and β-ZEL were determined as the concentration at which the signal-to-noise ratio decreased to 3. The concentrations and percentage content of ZEN, α-ZEL and β-ZEL were determined in each group and on both analytical dates (Table 1). 

##### Carry-Over Factor

Carry-over toxicity takes place when organisms exposed to low doses of mycotoxins survive. Mycotoxins can compromise tissue or organ functions [59] and modify their biological activity [4,15]. The carry-over factor (CF) was determined in the myocardium when the daily dose of ZEN (5 µg ZEN/kg BW, 10 µg ZEN/kg BW or 15 µg ZEN/kg BW) administered to each animal was equivalent to 560–6255 µg ZEN/kg of the complete diet, depending on daily feed intake. Mycotoxin concentrations in tissues were expressed in terms of the dry matter content of the samples. 

The CF was calculated as follows:(1)$$carry-over\;factor=\frac{toxin\;concentration\;in\;tissue\;\left[\mathrm{ng}/g\right]}{toxin\;concentration\;in\;diet\;\left[\mathrm{ng}/g\right]}$$

##### Statistical Analysis 

Data were processed statistically at the Department of Discrete Mathematics and Theoretical Computer Science, Faculty of Mathematics and Computer Science of the University of Warmia and Mazury in Olsztyn. The bioavailability of ZEN and its metabolites in the apex of heart was analyzed in group C and three experimental groups on two analytical dates. The results were expressed as means ($\bar{x}$) with standard deviation (SD). The following parameters were analyzed: (i) differences in the mean values for three ZEN doses (experimental groups) and the control group on both analytical dates, and (ii) differences in the mean values for specific ZEN doses (groups) on both analytical dates. In both cases, the differences between mean values were determined by one-way ANOVA. If significant differences were noted between groups, the differences between paired means were determined by Tukey’s multiple comparison test. If all values were below LOD (mean and variance equal zero) in any group, the values in the remaining groups were analyzed by one-way ANOVA (if the number of the remaining groups was higher than two), and the means in these groups were compared against zero by Student’s t-test. Differences between groups were determined by Student’s *t*-test. The results were regarded as highly significant at *P* < 0.01 (**) and as significant at 0.01 < *P* < 0.05 (*). Data were processed statistically using Statistica v.13 software (TIBCO Software Inc., Silicon Valley, CA, USA, 2017).

### 4.2. Laboratory Analyses

#### 4.2.1. Sampling for In Vitro Analyses

Tissue samples for in vitro analyses were collected from the same animals as the tissues for the toxicological analysis. The animals were euthanized by intravenous administration of pentobarbital sodium (Fatro, Ozzano Emilia BO, Italy) and exsanguinated. 

#### 4.2.2. In Vitro Analysis

In vitro analyses were performed at the Department of Pharmacology and Toxicology of the Faculty of Medicine at the University of Warmia and Mazury in Olsztyn, Poland. 

#### 4.2.3. Drugs

Acetylcholine (ACh) chloride, sodium nitroprusside (SNP), bradykinin acetate salt, indomethacin, N^G^-nitro-L-arginine methyl ester (L-NAME) and potassium chloride (KCl) (Chempur, Poland) were obtained from Sigma-Aldrich unless stated otherwise. Stock solutions (10 mM) of indomethacin and L-NAME were prepared in DMSO. These solutions were stored at −20 °C, and appropriate dilutions were made in Krebs-Henseleit solution (KHS in mM: NaCl 120, KCl 4.76, NaHCO_3_ 25, NaH_2_PO_4_·H_2_O 1.18, CaCl_2_ 1.25, MgSO_4_·7H_2_O 1.18, glucose 5.5) on the day of the experiment. At these concentrations, DMSO did not alter the reactivity of coronary arteries.

#### 4.2.4. Vascular Reactivity Analyses 

Porcine hearts were transported on ice and rinsed with cold aerated KHS. The left anterior descending branches of the coronary artery (MID LAD, D1-D2 sections) were excised at room temperature (*n* = 5). The surrounding connective tissues were removed, the arteries were cut into 4 mm rings and suspended between two stainless-steel rods positioned in 5 mL tissue baths (Graz Tissue Bath System, Harvard Apparatus) filled with aerated (95% O_2_ + 5% CO_2_) KHS at 37 °C (pH 7.4). The resting tension of 1.5 g was applied and further readjusted every 15 min during a 60 min equilibration period before further analysis. 

The viability of each porcine coronary artery was determined by contraction with 30 mM KCl before relaxation with 1 mM bradykinin. Rings that failed to produce an average contraction greater than or equal to 4.0 g when challenged with KCl and relaxation greater than or equal to 40% when treated with bradykinin were excluded from the study (approx. 10–20% of prepared rings). The rings were washed three times with fresh KHS, and baseline tension was readjusted before the examination.

After an initial equilibration period of 60 min, contractile responses elicited by either a cumulative concentration of KCl (2.5–30 mM) or a single maximum depolarizing concentration of KCl (30 mM) were assessed. The rings were also assessed for contractile responses generated by a cumulative concentration of acetylcholine (ACh: 0.1–10 µM). The cumulative concentrations of sodium nitroprusside (SNP: 0.1–100 µM) were assessed to determine the possible contribution of exogenous nitric oxide (NO) to relaxation responses on endothelium-intact rings that had been precontracted with submaximal concentrations of KCl (30 mM). 

In experiments evaluating the influence of COX and e-NOS-inhibitors on contractile responses, indomethacin (4.4 × 10^−6^ M) and L-NAME (4.4 × 10^−5^ M) were added to the chambers 30 min before the arteries were constricted with acetylcholine. Each tissue was exposed to the contracting agent only once.

#### 4.2.5. Statistical Analysis

Data were expressed as means ± SD (Standard Deviation), where *n* denoted the number of porcine hearts from which arterial rings were obtained. The contraction elicited by KCl was expressed in g of tension. ACh-induced contraction and SNP-induced relaxation were expressed as a percentage of the initial contraction elicited by 30 mM KCl. Dose-response curves were analyzed for the area under the curve, E_max_ and EC_50_ in GraphPad Prism 9.0.2. Data were processed statistically by comparing the curves obtained in each experimental group with the control curve in two-way ANOVA, followed by Tukey’s multiple comparisons test. Differences were regarded as significant at *P* ≤ 0.05.

## Figures and Tables

**Figure 1 toxins-13-00396-f001:**
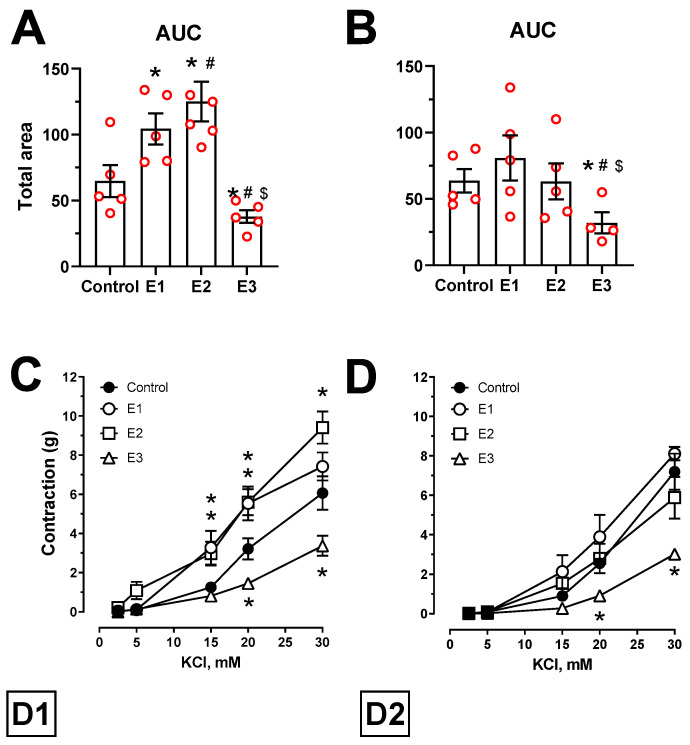
The effect of varying doses of ZEN (E1 = 5, E2 = 10, E3 = 15 μg ZEN/kg BW) on the cumulative contraction to KCl (2.5 to 30 mM) on exposure days D1 (7th day) (**A**,**C**) and D2 (21st day) (**B**,**D**). The results (means ± SEM) are expressed as AUC (**A**,**B**) and as a cumulative concentration-response curve in as grams of tension of porcine coronary arteries (**C**,**D**). *n* = 5. * *P <* 0.05 vs. control, ^#^
*P <* 0.05 vs. E1, ^$^
*P <* 0.05 vs. E2 (two-way ANOVA, followed by Tukey’s post-hoc test). Decreased contractile-response to KCl was observed in group E3 figure in D1 and D2 only. The effect of E1 and E2 was temporary and was limited to a shorter exposure (D1).

**Figure 2 toxins-13-00396-f002:**
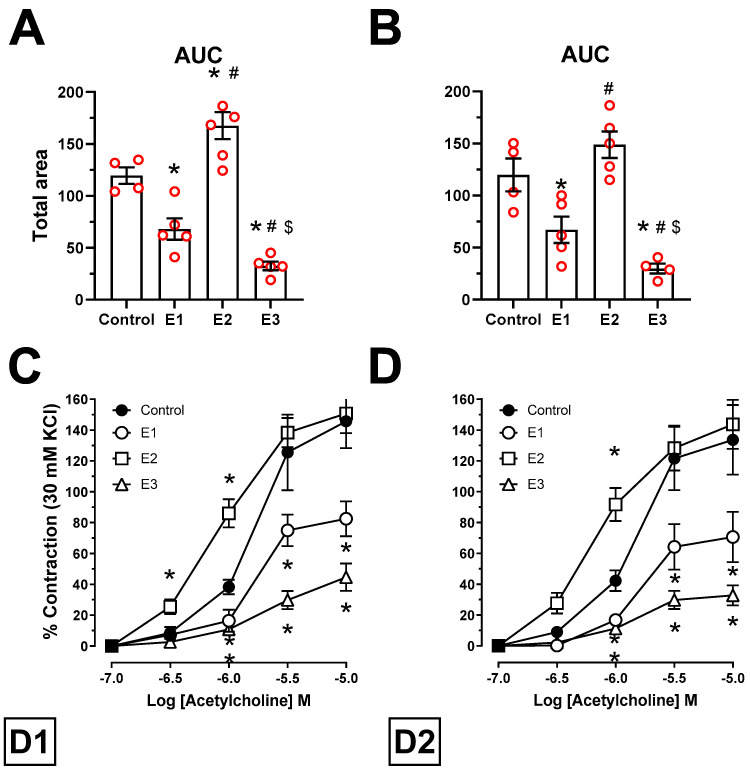
The effect of varying doses of ZEN (E1 = 5, E2 = 10, E3 = 15 μg ZEN/kg BW) on the cumulative contractions to acetylcholine on exposure day D1 (7th day) (**A**,**C**) and D2 (21st day) (**B**,**D**). The results (means ± SEM) are expressed as AUC (**A**,**B**) and as a cumulative concentration-response curve of the percentage inhibition of the contraction induced by 30 mM KCl of porcine coronary arteries (**C**,**D**). *n* = 5. * *P <* 0.05 vs. control, ^#^
*P <* 0.05 vs. E1, ^$^
*P <* 0.05 vs. E2 (two-way ANOVA, followed by Tukey’s post-hoc test). Both E1 and E3 decreased the contractile-response to acetylcholine in D1 and in D2. E2 modulated the response in D1 and this was not observed after longer exposure (in D2).

**Figure 3 toxins-13-00396-f003:**
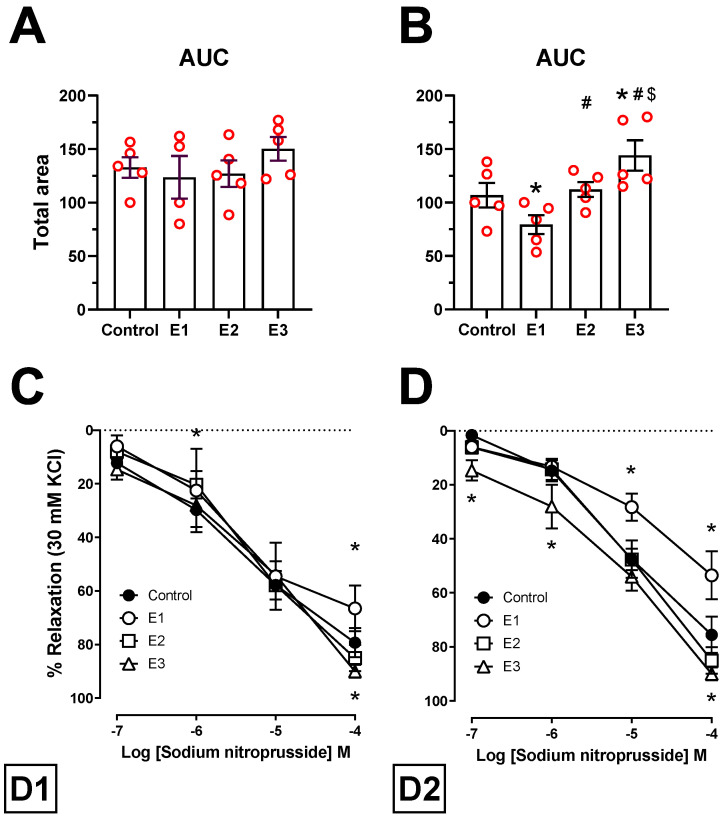
The effect of different doses of ZEN (E1 = 5, E2 = 10, E3 = 15 μg ZEN/kg BW) on the cumulative contractions to sodium nitroprusside (SNP) on exposure day D1 (7th day) (**A**,**C**) and D2 (21st day) (**B**,**D**). The results (means ± SEM) are expressed as AUC (**A**,**B**) and as a cumulative concentration-response curve of the percentage inhibition of the contraction induced by 30 mM KCl of porcine coronary arteries (**C**,**D**). *n* = 5. * *P <* 0.05 vs. control, ^#^
*P <* 0.05 vs. E1, ^$^
*P <* 0.05 vs. E2 (two-way ANOVA, followed by Tukey’s post-hoc test). An enhanced relaxant-response was observed in the E3 group in D2 only. In the E1 group decreased response was observed in term of longer exposure (D2).

**Figure 4 toxins-13-00396-f004:**
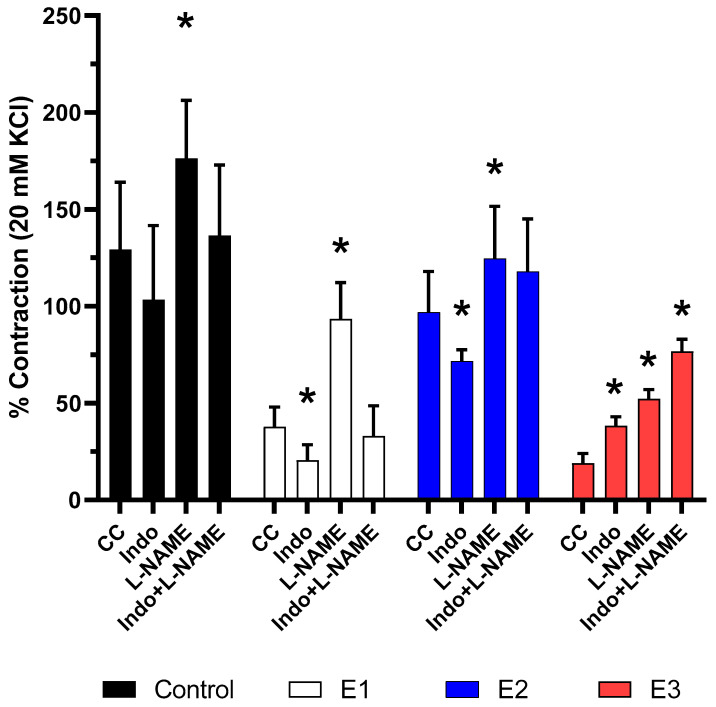
The effect of nitric oxide synthase inhibitor (L-NAME, 4.4 × 10^−5^ M), cyclooxygenase inhibitor (indomethacin, 4.4 × 10^−6^ M) on acetylcholine-induced contraction of the porcine coronary arteries. The results (means ± SEM) are expressed as percentage inhibition of the contraction induced by 30 mM KCl. *n* = 5. * *P <* 0.05 vs. control conditions (CC), (two-way ANOVA, followed by Tukey’s post-hoc test). Acetylcholine-induced concentration was potentiated in the presence of both Indo+L-NAME in the E3 group, but not in the E1 group and E2 group.

**Table 1 toxins-13-00396-t001:** The carry-over factor and the mean ($\bar{x}$) concentrations of ZEN and its metabolites (α-ZEL and β-ZEL) (ng/g) in the myocardium of prepubertal gilts.

Weeks of Exposure	Feed Intake [kg/day]	Total doses of ZEN in Groups Respectively [µg/kg BW]	Group E1 [ng/g] (%)	Carry-over Factor	Group E2 [ng/g]/(%)	Carry-over Factor	Group E3 [ng/g]/(%)	Carry-over Factor
Zearalenone								
D1	0.8	80.5/161.9/242.7	0.691 ± 0.635 (100%)	8 × 10^−6^	2.275 ± 2.22 (86.69%)	14 × 10^−6^	1.387 ± 1.93(78.22%)	17 × 10^−6^
D2	1.1	101.01/196.9/298.2	0.977 ± 0.579 (73.62%)	9 × 10^−6^	2.621 ± 1.499 (91.48%)	13 × 10^−6^	4.89 ± 4.405(93.42%)	16 × 10^−6^
α-ZEL								
D1	not applicable	not applicable	0.0 ± 0.0 (0%)	0	0.316 ± 0.061 (12.04%)	19 × 10^−7^	0.353 ± 0.104(19.9%)	14 × 10^−7^
D2	not applicable	not applicable	0.146 ± 0.143 (11%)	14 × 10^−7^	0.167 ± 0.146 (5.82%)	1 × 10^−6^	0.312 ± 0.213(5.96%)	1 × 10^−6^
β-ZEL								
D1	not applicable	not applicable	0.0 ± 0.0 (0%)	0	0.033 ± 0.029 (1.25%)	2 × 10^−7^	0.033 ± 0.004(1.86%)	1 × 10^−7^
D2	not applicable	not applicable	0.204 ± 0.046 (15.37%)	2 × 10^−6^	0.077 ± 0.017 ******(2.68%)	4 × 10^−7^	0.032 ± 0.004 ******(0.61%)	1 × 10^−7^

Abbreviation: D1—exposure day 7; D2—exposure day 21. Experimental groups: Group E1—5 μg ZEN/kg BW; Group E2—10 μg ZEN/kg BW; Group E3—15 μg ZEN/kg BW. LOD > values below the limit of detection were expressed as 0. The results were regarded as highly significant at *P* < 0.01 (**).

**Table 2 toxins-13-00396-t002:** Changes in vasoconstriction induced by acetylcholine and changes in vasodilatation induced by sodium nitroprusside (%) in porcine coronary arteries (expressed by AUC, E_max_ and pD_2_ values).

	Control			Group E1			Group E2			Group E3		
	AUC	E_max_ (%)	pD_2_	AUC	E_max_ (%)	pD_2_	AUC	E_max_ (%)	pD_2_	AUC	E_max_ (%)	pD_2_
**_D1_** **ACh**	122.6 ± 15.01	147.8 ± 11.33	5.801 ± 0.077	69.93 ± 11.20*	82.87 ± 7.853*	5.788 ± 0.103	162.6 ± 14.58*	155.1 ± 10.69	6.087 ± 0.074	32.73 ± 6.205*	51.08 ± 13.19*	5.606 ± 0.210
**_D2_** **ACh**	119.8 ± 19.10	135 ± 15.39	5.859 ± 0.112	58.29 ± 13.33*	70.89 ± 10.28*	5.831 ± 0.149	137.6 ± 9.264	111.3 ± 10.30	6.263 ± 0.101*	29.81 ± 5.434*	33.13 ± 4.47*	5.876 ± 0.129
**_D1_** **SNP**	137.5 ± 10.36	76.9 ± 5.03	5.469 ± 0.2543	116.3 ± 21.18	67.74 ± 9.994	5.605 ± 0.433	128.6 ± 14.23	89.5 ± 5.012	5.217 ± 0.153	134.3 ± 12.20	97.78 ± 7.674	4.952 ± 0.175
**_D2_** **SNP**	101.6 ± 18.32	79.99 ± 6.529	5.171 ± 0.175	74.25 ± 11.65*	60.12 ± 8.205*	4.833 ± 0.273	110.5 ± 13.74	93.53 ± 5.480	4.963 ± 0.116	141.7 ± 12.62*	97.78 ± 7.674*	4.952 ± 0.175

Abbreviations: AUC, area under the dose-response curve; E_max_, maximal response values; pD_2_, drug concentration exhibiting 50% of the Emax expressed as the negative log molar; SNP, sodium nitroprusside. Values are expressed as mean ± S.E.M. **P* < 0.05 vs. the control group (one-way ANOVA, followed by Tukey’s post-hoc test).

**Table 3 toxins-13-00396-t003:** Declared composition of the complete diet.

Ingredient	Manufacturer’s Declared Composition (%)
Soybean meal	16
Wheat	55
Barley	22
Wheat bran	4.0
Chalk	0.3
Zitrosan	0.2
Vitamin-mineral premix ^1^	2.5

^1^ Composition of the vitamin-mineral premix per kg: vitamin A—500,000 IU; iron—5000 mg; vitamin D3—100,000 IU; zinc—5000 mg; vitamin E (alpha-tocopherol)—2000 mg; manganese—3000 mg; vitamin K—150 mg; copper (CuSO_4_·5H_2_O)—500 mg; vitamin B_1_—100 mg; cobalt—20 mg; vitamin B—300 mg; iodine—40 mg; vitamin B_6_—150 mg; selenium—15 mg- vitamin B_12_—1500 μg; L-lysine—9.4 g; niacin—1200 mg; DL-methionine + cystine—3.7 g; pantothenic acid—600 mg; L-threonine—2.3 g; folic acid—50 mg; tryptophan—1.1 g; biotin—7500 μg; phytase + choline—10 g; ToyoCerin probiotic+calcium—250 g; antioxidant+mineral phosphorus and released phosphorus—60 g; magnesium—5 g; sodium and calcium—51 g.

## Data Availability

Not applicable.

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
