# Peer review of "Concentration of Zearalenone, Alpha-Zearalenol and Beta-Zearalenol in the Myocardium and the Results of Isometric Analyses of the Coronary Artery in Prepubertal Gilts"

_toxins, 2021, doi:10.3390/toxins13060396_

Round 1

Reviewer 1 Report

Introduction:

The introduction could use a little more detail. For instance, line 41 what clinical symptoms? The reader should not have to go read other papers to determine what it is you are discussing.

Results:

Line128: You reference table 1 to show AUC for D2. Table 1 doesn't not show this. I think you meant table 2 and figure 1. 

Line147: what is meant by conTable 1?

Figures 1-3: I would suggest sub-panel labels in the figures (i.e. Figure 1A, Figure 1B, etc. The figures are not intuitive based on your results section. I would also suggest more detail. What are the major results to take away from each figure? 

Table 2 is not mentioned in the results. If it is going to be included, it needs to be mentioned in the results. 

Line 213-215: Provide more detail. I do not get that conclusion from your presented data.

Author Response

Thanks for your comments. The suggestions presented are very beneficial to the proposed article.

Introduction:

The introduction could use a little more detail. For instance, line 41 what clinical symptoms? The reader should not have to go read other papers to determine what it is you are discussing.

Response - Basic clinical conditions have been presented and supported by relevant publications.

Results:

Line128: You reference table 1 to show AUC for D2. Table 1 doesn't not show this. I think you meant table 2 and figure 1. 

Response - Yes, it should be Table 2 and Figure 1. The error has been corrected.

 Line147: what is meant by conTable 1?

Response - The indicated stylistic error has been corrected.

Figures 1-3: I would suggest sub-panel labels in the figures (i.e. Figure 1A, Figure 1B, etc. The figures are not intuitive based on your results section. I would also suggest more detail. What are the major results to take away from each figure? 

Response - All comments and suggestions have been made.

Table 2 is not mentioned in the results. If it is going to be included, it needs to be mentioned in the results. 

Response - The error has been completed

Line 213-215: Provide more detail. I do not get that conclusion from your presented data.

Response - We expect that the arguments presented in the text will satisfy the Reviewer.

Reviewer 2 Report

The manuscript dealt with the effect of low ZEN doses to prepubertal gilts on the accumulation of ZEN, α-ZEL and β-ZEL in the myocardium and the reactivity of the left anterior descending branch of the coronary artery to acetylcholine, 9 potassium chloride, and sodium nitroprusside. The data has to be revisited and need redrawing of the conclusion. Please find my comments below:

1) Most of the data in figures seems to curve fitted and smoothed, which shouldn't be the case if the plot is used for AUC calculation. The author should change that and recalculate AUC. All the data could be presented as SEM.

2) The AUC of E2 in Fig1 D2 seems to have quite a bigger SD and the stat looks a bit suspicious in comparison to E1. The author should present the AUC in the individual data point form with SEM (apply to all the AUC plots). 

3) In figure 2D2, some of the points of E2 seem to be off, maybe because of curve fitting, it would be interesting to see without fitting replot of the AUC.

4) In figure 3D2, the calculated AUC doesn't follow the graph as E3 looks like will be the lowest in comparison to all others, however presented higher than others.

5) Why looks at the effect of NOS inhibition rather than ROS? It would be interesting to see the ROS status in the myocardium?

6) Minor: all the graphs should have their own labeling (all D2). Figure 4, ACh 10^-6 M is unnecessary under the figure.

Author Response

Thank you very much for your comments. The reviewers found a few mistakes that we could not avoid while preparing the manuscript. We appreciate this as it would be unfavorable to the manuscript. We have revised the manuscript following the comments below.

The manuscript dealt with the effect of low ZEN doses to prepubertal gilts on the accumulation of ZEN, α-ZEL and β-ZEL in the myocardium and the reactivity of the left anterior descending branch of the coronary artery to acetylcholine, 9 potassium chloride, and sodium nitroprusside. The data has to be revisited and need redrawing of the conclusion. Please find my comments below:

1) Most of the data in figures seems to curve fitted and smoothed, which shouldn't be the case if the plot is used for AUC calculation. The author should change that and recalculate AUC. All the data could be presented as SEM.

Response - In the first version of the manuscript, AUC was calculated before curve fitting. The curves were then developed. We have now removed the curve fit and moved that into additional data. The data presented was SEM and not SD, in a few cases it was both SEM and SD, which has been adjusted. We also corrected the captions for figs.

2) The AUC of E2 in Fig1 D2 seems to have quite a bigger SD and the stat looks a bit suspicious in comparison to E1. The author should present the AUC in the individual data point form with SEM (apply to all the AUC plots). 

Response - We have corrected accordingly.

3) In figure 2D2, some of the points of E2 seem to be off, maybe because of curve fitting, it would be interesting to see without fitting replot of the AUC.

Response - We removed curve fitting.

4) In figure 3D2, the calculated AUC doesn't follow the graph as E3 looks like will be the lowest in comparison to all others, however presented higher than others.

Response - The AUC was calculated to baseline 0 not to 100, so E1 is the smallest and E3 is the highest. We have modified a graph and a baseline was added.

5) Why looks at the effect of NOS inhibition rather than ROS? It would be interesting to see the ROS status in the myocardium?

Response - This was a pilot study to check if there is any response to supplementation. Further studies will focus on ROS and will study other mechanisms.

6) Minor: all the graphs should have their own labeling (all D2). Figure 4, ACh 10^-6 M is unnecessary under the figure.

Response - It was corrected

Reviewer 3 Report

The authors have shown the action of ZEN and its metabolites on coronary artery reactivity by using prepubertal gilts.  The experimental design and the results seem to be reasonable, however, the significancy of the aim is too vague to be useful for the readers. The results are phenomenological and the mechanistic insight are lacking although they examined by using L-NAME and indomethacin. In addition, they did not check myocardium at all, which is different from the title. The manuscript is unfortunately unsuitable for the publication of in toxins.

Author Response

The authors have shown the action of ZEN and its metabolites on coronary artery reactivity by using prepubertal gilts.  The experimental design and the results seem to be reasonable, however, the significancy of the aim is too vague to be useful for the readers. The results are phenomenological and the mechanistic insight are lacking although they examined by using L-NAME and indomethacin. In addition, they did not check myocardium at all, which is different from the title. The manuscript is unfortunately unsuitable for the publication of in toxins.

Response - We don't understand the arguments used by the Reviewer to suggest that "The manuscript is unfortunately unsuitable for the publication of in toxins."

When doing any mycotoxicological tests, it is better to do preliminary tests that will give you a place for a specific hypothesis. The aim of this study was to check whether very low doses of ZEN administered per os reach the myocardium and whether induces changes in reactivity in response to vasoconstrictors and vasodilators in the left anterior descending branch of the coronary artery. For clarity, we propose to slightly change the title of the work, which will make it compatible with the aims and summary of the work.

Round 2

Reviewer 2 Report

The authors have addressed all my concerns.

Reviewer 3 Report

The present study is out of my research area, and it is difficult to understand the significance of aim in the study.

If possible, please ask another reviewer (I recommend Veterinarian) before giving final decision.